# Exploring the limits of exercise capacity in adults with type II diabetes

**Matthijs Michielsen**[1,2☯], **Youri Bekhuis**[3,4,5☯], **Jomme Claes**[1], **Elise Decorte**[1], **Camille De Wilde**[1], **Tin Gojevic**[6,7], **Louise Costalunga**[1], **Sara Amyay**[1], **Varvara Lazarou**[1], **Daphni Daraki**[1], **Eleftheria Kounalaki**[1], **Panagiotis Chatzinikolaou**[8], **Kaatje Goetschalckx**[4], **Dominique Hansen**[6,7], **Guido Claessen**[3,5,9], **Marieke De Craemer**[2], **Véronique Cornelissen**[1*]

1 Department of Rehabilitation Sciences, KU Leuven, Leuven, Belgium, 2 Department of Rehabilitation Sciences, Ghent University, Gent, Belgium, 3 Department of Cardiovascular Sciences, KU Leuven, Leuven, Belgium, 4 Department of Cardiovascular Diseases, UZ Leuven, Leuven, Belgium, 5 Faculty of Medicine and Life Sciences (LCRC), Hasselt University, Diepenbeek, Belgium, 6 Faculty of Rehabilitation Sciences, Hasselt University, Hasselt, Belgium, 7 Rehabilitation Research Center (REVAL) and Biomedical Research Institute (BIOMED), Hasselt University, Diepenbeek, Belgium, 8 Department of Physical Education and Sports Sciences at Serres, Aristotle University of Thessaloniki, Thessaloniki, Greece, 9 Department of Cardiology, Jessa Hospital, Hasselt, Belgium

☯ These authors have contributed equally to this work.
* matthijs.michielsen@kuleuven.be

## Abstract

### Objective

This study investigates the mechanisms behind exercise capacity in adults with type 2 diabetes mellitus (T2DM), focusing on central and peripheral components, as described by the Fick equation.

### Methods

A cross-sectional study of 141 adults with T2DM was conducted, using cardiopulmonary exercise testing, near-infrared spectroscopy (NIRS) and exercise echocardiography. Participants with sufficient-quality NIRS data were stratified into tertiles based on percentage predicted $VO_2$peak. Group comparisons and stepwise regression were used to examine the contributions of central and peripheral components to $VO_2$peak.

### Results

Sixty-seven participants had insufficient quality NIRS data. Those with lower-quality data were more likely to be female ($p < 0.001$) and had a lower exercise capacity ($p < 0.001$). Among participants with good-quality NIRS data, those in the lowest fitness tertile were older ($p < 0.01$), had a longer diabetes duration ($p = 0.04$), lower eGFR ($p < 0.001$) and more frequent use of beta-blockers ($p = 0.02$) and diuretics ($p = 0.04$). Significant differences were observed in peak cardiac output ($p < 0.001$) and NIRS-derived parameters across fitness groups. Multivariate regression

**Data availability statement:** All relevant data are within the manuscript and its Supporting Information files.

**Funding:** This trial received funding from the Scientific Research Foundation of Flanders (FWO – T004420N and FWO – G095221N). The funders had no role in study design, data collection and analysis, decision to publish, or preparation of the manuscript.

**Competing interests:** The authors have declared that no competing interests exist.

identified cardiac output as the strongest predictor of $VO_2$peak, while peripheral oxygen extraction did not improve model performance.

## Conclusion

Cardiac output is the primary determinant of exercise capacity in adults with T2DM. This suggests that muscle perfusion may be the main limiting factor in relatively fit individuals with T2DM. However, cardiac output and local muscle perfusion are not directly equivalent, as mechanical factors, such as intramuscular pressure during high-intensity exercise, may prevent maximal perfusion.

---

## Introduction

Adults with type 2 diabetes (T2DM) often present with a significantly reduced exercise capacity, demonstrated by a 20–30% lower peak oxygen consumption ($VO_2$peak) compared to their healthy peers [1,2]. This reduced $VO_2$peak is a key factor contributing to adverse clinical outcomes and reduced life expectancy in this population [3,4]. An improvement in $VO_2$peak by one metabolic equivalent of a task (MET) is associated with a 14–19% reduction in mortality risk [5,6]. However, exercise capacity varies widely among adults with T2DM [1]. Therefore, a better understanding of the underlying mechanisms associated with a lower exercise capacity in adults with T2DM is needed to facilitate early preventive interventions.

According to the Fick equation, $VO_2$peak is the product of cardiac output (CO) and the arteriovenous oxygen difference (a-v $O_2$ diff), representing the central and peripheral components of oxygen transport, respectively [7]. The central component, CO, can be reliably measured using echocardiography, a non-invasive and widely used imaging technique [8,9]. However, evidence regarding the role of CO in exercise intolerance among adults with T2DM remains inconclusive. While some studies have found no differences in CO, others suggest that impaired CO adjustment during exercise is a key factor in reduced exercise capacity [10–12].

Peripheral oxygen extraction by the muscle is also often recognized as a key factor contributing to exercise intolerance in adults with T2DM [10,11]. In particular, insulin resistance is closely linked to mitochondrial dysfunction, leading to reduced respiration rates and impaired substrate utilization [13,14]. Additionally, T2DM is associated with increased arterial stiffness and endothelial dysfunction, which may compromise both oxygen delivery and extraction in the working muscles [15,16]. Near-infrared spectroscopy (NIRS) is a non-invasive method for assessing oxygen-dependent absorption of oxygenated hemoglobin ($O_2$Hb) and deoxygenated hemoglobin (HHb) in muscle tissue [17]. NIRS has been successfully used in various populations to evaluate muscle oxygenation and microvascular reactivity, making it an interesting tool for evaluating the peripheral contribution to exercise capacity in adults with T2DM [18–20].

In summary, the underlying mechanisms contributing to reduced exercise capacity in adults with T2DM remain inconclusive. Specifically, we aim to determine whether

differences in $VO_2$peak between people with T2DM are primarily driven by differences in cardiac output, peripheral oxygen extraction or a combination of both.

## Materials and methods

### Study design and participants

This cross-sectional study included baseline data from participants with T2DM enrolled in two exercise intervention trials in UZ/KU Leuven (Belgium) (PROTECTION trial-NCT05023538, recruitment between 28/02/2022 and 07/05/2024 | PRIORITY trial-NCT04745013, recruitment between 16/09/2021 and 29/03/2024). Both study protocols adhered to the Declaration of Helsinki and were approved by the Ethics Committee Research UZ/KU Leuven. Before enrolment, all participants provided written informed consent. Eligibility criteria for this study included adults aged 35–80 years with a diagnosis of T2DM and on stable pharmacological therapy for at least 4 weeks. Exclusion criteria included participants with uncontrolled diabetes (HbA1c > 9%), uncontrolled hypertension, significant arrhythmias, established cardiovascular disease, chronic obstructive pulmonary disease, cerebrovascular, renal or peripheral vascular disease and active malignant disease.

### Measurements

**Clinical characteristics.** Medication use was assessed verbally, while demographic data, smoking status, and medical history were collected by a questionnaire and verified in medical records. Fasted blood samples were taken to measure fasting plasma glucose (FPG), hemoglobin A1c (HbA1c), hemoglobin (Hb), total cholesterol, low-density lipoprotein cholesterol (LDL), high-density lipoprotein cholesterol (HDL), triglycerides, creatinine and estimated glomerular filtration rate (eGFR). Blood pressure was measured in triplicate (Omron X3, Omron Healthcare, Japan). The percentage body fat was measured using the Bodystat Quadscan 4000 (Bodystat Ltd, British Isles). Body height and body mass (Seca Alpha 770, Seca, Germany) were measured barefoot and body mass index (BMI) was calculated as body mass in kilograms divided by height in meters squared ($kg/m^2$). Waist circumference was measured twice at the level of the umbilicus.

**Cardiopulmonary exercise test (CPET).** All participants performed a symptom-limited graded cardiopulmonary exercise test (CPET) on a cycle ergometer (Vyntus CPX, Duomed, Belgium) until volitional exhaustion (i.e., when participants were no longer able to maintain a cycling frequency of 60 rpm), followed by a three-minute recovery period of unloaded pedaling. An individualized ramp protocol was applied, starting at either 10, 20 or 50 watts, with respective increments of 10, 20 or 25 watts per minute, depending on participants' physical status. This approach aimed to achieve a total test duration between 8–12 minutes [21]. Heart rate and a 12-lead electrocardiogram (CardioSoft ECG, CardioSoft, USA) were recorded continuously. Blood pressure (SunTech Tango M2, SunTech Medical, USA) was measured automatically every other minute. Additionally, a breath-by-breath analysis of ventilation and pulmonary gas exchange parameters was performed (SentrySuite, Duomed, Belgium). Ratings of perceived exertion (Borg scale) at the end of the test and reasons for stopping the exercise test (i.e., muscle fatigue and/or shortness of breath) were noted. $VO_2$peak was determined as the highest average oxygen uptake over 30 seconds.

**Near-infrared spectroscopy (NIRS).** Quadriceps muscle oxygenation was measured with a wireless continuous-wave three-channel NIRS device (PortaMon, Artinis, The Netherlands) 15 centimeter proximal to the lateral femoral epicondyle on the mid-portion of the vastus lateralis muscle of the right leg. The Beer-Lambert Law was used to calculate changes in tissue saturation index (TSI), oxygenated hemoglobin ($O_2$Hb) and deoxygenated hemoglobin (HHb). Total hemoglobin (tHb) was calculated as the sum of $O_2$Hb and HHb and hemoglobin difference ($Hb_{diff}$) was calculated as $O_2$Hb minus HHb. To minimize noise, data were down-sampled to 1 Hz and a moving Gaussian filter with a 3-second window was applied. Variations in $O_2$Hb, HHb, tHb, and $Hb_{diff}$ were expressed as an average of the 3 optodes and were normalized to reflect changes from baseline level. NIRS parameters during exercise were reported at percentages of $VO_2$peak. During

recovery, NIRS parameters were reported at 10-second intervals for the first minute post-exercise, followed by 30-second intervals until the end of the recovery period. To quantify the changes in NIRS parameters, the differences between the maximum and minimum values were calculated. In addition, the ΔHHb/ΔtHb ratio was calculated as an index of oxygen extraction relative to local blood volume. Only NIRS measurements of sufficient quality were included, as determined by the absence of a flat line or TSI fit factors exceeding 99%.

**Echocardiography at rest and during exercise.** Resting and exercise echocardiography, combined with CPET (CPETecho), were performed using a Vivid E95 ultrasound system (GE Healthcare, USA), one week after the standard CPET assessment. The imaging protocol for resting echocardiography included the measurements of conventional morphological parameters and CO. CO was calculated using the velocity-time integral of the left ventricular outflow tract (LVOT) obtained via pulsed-wave Doppler, along with heart rate and LVOT diameter. The standardized CPETecho protocol, previously described in detail [22], was conducted on a semi-supine bicycle ergometer (Ergoline, GmbH, Bitz, Germany) using an individualized ramp protocol. Images were acquired before exercise, at low intensity (heart rate between 90 and 100 beats per minute, before fusion of E and A waves, or at a respiratory exchange ratio (RER) between 0.85 and 0.9 in case of chronotropic incompetence), and at peak exercise (RER > 1.05). All analyses were performed offline using EchoPAC software (version 204, GE Vingmed, Norway) in accordance with contemporary international guidelines [23,24].

## Statistical analysis

All statistical analyses were conducted using JASP Statistics (version 0.19.1, JASP Stats, The Netherlands). Participants were first categorized into two groups based on NIRS data quality: high-quality and insufficient quality NIRS data. Subsequently, participants with high-quality NIRS data were divided into tertiles, based on the predicted $VO_2$peak (Gläser et al., 2010) [25]. Characteristics of participants in the lowest (tertile 1) and highest (tertile 3) fitness group were compared. Data normality was assessed through visual inspection of Q-Q plots and histograms. Normally and non-normally distributed variables are presented as mean ± standard deviation (SD) and median ± interquartile range (IQR), respectively. Group differences were analyzed using an independent t-test (normally distributed data) or a Mann-Whitney U test (non-normally distributed data). To evaluate the potential confounding effect of adipose tissue on NIRS outcomes, analysis of covariance (ANCOVA) was performed with adipose tissue thickness (ATT) as a covariate. As ATT could not be measured in all participants, a parallel ANCOVA was conducted using body fat percentage as an alternative covariate. To assess differences in the temporal evolution of NIRS parameters during recovery, a repeated measures analysis of variance (ANOVA) was conducted comparing the lowest and highest fitness tertiles. Mauchly's test of sphericity was applied to assess homogeneity of variance, and, where violated, Greenhouse–Geisser corrections were applied to the degrees of freedom. Post hoc comparisons were performed using Holm's correction to control for multiple testing. Finally, a stepwise multiple regression analysis was conducted to assess the individual contributions of central and peripheral factors to $VO_2$peak (ml/min). The baseline model included the variables described by Gläser et al. 2010 as independent variables (age, gender, height, weight and smoking status) [25]. CO (central component) and the ΔHHb/ΔtHb ratio (peripheral component) were then added separately to evaluate their additional explanatory value. Model performance was evaluated using adjusted $R^2$, Bayesian Information Criterion (BIC) values and root mean square error (RMSE).

## Results

### High vs insufficient quality measurement

A total of 141 participants with T2DM (79 men; mean age 61.41 ± 10.38 years old) performed a CPET combined with NIRS. Data from 67 participants (48%) were excluded from further analysis due to the insufficient quality of the NIRS measurements. A detailed comparison between participants with high-quality vs insufficient-quality NIRS is provided in Supplementary File S1 Table. Overall, participants with insufficient quality NIRS data were more likely to be female (p < 0.001),

had a higher fat mass (p < 0.001) and had a lower exercise capacity (p < 0.001) compared to those with high-quality NIRS measurements.

### Lowest vs highest fitness

The remaining 74 participants were categorized into tertiles based on their percentage of predicted $VO_2$peak [25]. Tertile 1 included 25 participants (22 men, average predicted $VO_2$peak = 77%) with the lowest fitness, while tertile 3 comprised 25 participants (20 men, average predicted $VO_2$peak = 118%) with the highest fitness.

**Demographics and clinical characteristics.** As shown in Table 1, participants in the lowest fitness tertile were on average older (p = 0.01), had a longer history of diabetes (p = 0.04) and had a worse kidney function, as indicated by a lower eGFR (p < 0.001). Furthermore, participants in the lowest fitness tertile had a higher fat mass (p < 0.004), a lower diastolic blood pressure (p = 0.04) and were more likely to use beta-blockers (p = 0.02) and diuretics (p = 0.04).

**Rest and exercise echocardiography.** As shown in Table 2, no significant differences were observed in resting echocardiography parameters between the fitness groups. However, at peak exercise, significant differences were found in CO (p < 0.001), cardiac index (CI) (p < 0.001) and peak heart rate (p < 0.001).

**Evolution of NIRS parameters during exercise.** Participants in the lowest fitness group exhibited a smaller increase in tHb (p < 0.001) during exercise, as well as a smaller change in $O_2$Hb (p < 0.001), tHb (p < 0.001) and $Hb_{diff}$ (p = 0.005) during the recovery period, compared to those in the higher fitness group. For HHb, a similar trend towards smaller increases during exercise and smaller decreases during recovery was observed, although these differences did not reach statistical significance (p = 0.07 for both). Results of the ANCOVA, adjusting for ATT and body fat are provided in Supplementary Files S2–S3 Tables. Both analyses yielded findings consistent with the original analysis and the observed statistical significance remained unchanged.

The evolution of NIRS-derived parameters during exercise and the recovery period is illustrated in Fig 1. A significant interaction effect between fitness group and time point was observed for changes in TSI (p = 0.04), $O_2$Hb (p < 0.001), and tHb (p = 0.001) during exercise and for $O_2$Hb (p < 0.001), tHb (p < 0.001) and $Hb_{diff}$ (p = 0.004) during recovery. Changes were consistently greater in the highest fitness group compared to the lowest fitness group. Post-hoc analyses indicated significant differences at 80% (p = 0.022) and 90% (p = 0.005) of $VO_2$peak for $O_2$Hb and at 70% (p = 0.004), 80% (p = 0.003), 90% (p < 0.001), and 100% (p = 0.001) of $VO_2$peak for tHb. During recovery, $O_2$Hb demonstrated significant differences from 20 seconds onward (p = 0.01), persisting across all subsequent time points until the end of the recovery period (p < 0.001). Likewise, tHb and $Hb_{diff}$ showed significant differences from 30 seconds onward (p = 0.04 and p = 0.01, respectively).

### Central vs peripheral contribution to $VO_2$peak

The results of the stepwise multiple regression analysis are presented in Table 3. The baseline model (Model 1), which included the covariates age, gender, height, weight and smoking status was significant (p < 0.001) and explained 53% of the variance in $VO_2$peak (adjusted $R^2$ = 0.53, RMSE = 483.41, BIC = 1151.85). The central component, Peak CO, showed a strong and significant (r = 0.63, p < 0.001) correlation with $VO_2$peak, which was reflected in improved model performance in Model 2 (adjusted $R^2$ = 0.61, RMSE = 436.21, BIC = 803.79). In contrast, the peripheral component, ΔHHb/ΔtHb, was not significantly associated with $VO_2$peak and provided minimal model improvement (adjusted $R^2$ = 0.54, RMSE = 475.58, BIC = 1149.43). The model including both peak CO and ΔHHb/ΔtHb (Model 5) achieved the best overall fit (adjusted $R^2$ = 0.62, RMSE = 429.79, BIC = 805.03), though its performance was comparable to the model including peak CO alone.

## Discussion

To our knowledge, this is the first study to examine the oxygen cascade, as defined by the Fick equation, by combining CO assessment and NIRS-derived skeletal muscle hemodynamics during exercise, within the same participants.

**Table 1. Demographics and clinical characteristics.**

| Based on Gläser (2010) | Total (N=74) | Lowest fitness (N=25) | Highest fitness (N=25) | p value |
|---|---|---|---|---|
| ***Demographics*** | | | | |
| Age (years) | 61.72±9.19 | 64.75±9.25 | 58.33±7.98 | **0.01** |
| Sex (M/F) | 62/12 | 20/5 | 22/3 | 0.44 |
| Duration of diabetes (years)* | 6.00±9.75 | 9.00±15.00 | 3.00±4.50 | **0.04** |
| ***Medication intake*** | | | | |
| Beta-blocker | 23 (31%) | 12 (48%) | 4 (16%) | **0.02** |
| Calcium channel blocker | 16 (22%) | 4 (16%) | 6 (24%) | 0.48 |
| Diuretics | 23 (31%) | 12 (48%) | 5 (20%) | **0.04** |
| Lipid-lowering drug | 49 (66%) | 18 (72%) | 13 (52%) | 0.15 |
| Metformin | 66 (89%) | 21 (84%) | 22 (88%) | 0.68 |
| Insulin | 9 (12%) | 3 (12%) | 2 (8%) | 0.64 |
| SGLT2-inhibitor | 17 (23%) | 6 (24%) | 6 (24%) | 1.00 |
| GLP1-agonist | 26 (35%) | 10 (40%) | 7 (28%) | 0.37 |
| Sulfamines | 13 (18%) | 7 (28%) | 3 (12%) | 0.16 |
| DPP4-inhibitor | 4 (5%) | 2 (8%) | 0 (0%) | 0.15 |
| ***Blood pressure*** | | | | |
| Resting SBP (mmHg) | 130.25±15.10 | 128.59±16.73 | 134.65±12.18 | 0.16 |
| Resting DBP (mmHg) | 82.06±10.73 | 78.69±11.08 | 85.13±9.60 | **0.04** |
| **Smoking status** | | | | |
| Non-smoker | 38 (51%) | 11 | 12 | 0.78 |
| Smoker | 27 (37%) | 9 | 12 | 0.39 |
| Ex-smoker | 9 (12%) | 5 | 1 | 0.08 |
| ***Anthropometrics*** | | | | |
| Body mass index | 29.68±5.17 | 29.58±5.05 | 30.15±5.48 | 0.70 |
| Body weight (kg) | 91.14±19.61 | 90.00±21.14 | 93.34±19.13 | 0.56 |
| Fat mass (%) | 31.02±6.21 | 33.50±5.79 | 28.28±6.35 | **0.004** |
| Waist circumference (cm) | 108.74±14.66 | 111.11±16.33 | 107.45±14.18 | 0.41 |
| Adipose tissue thickness (mm) – Vastus lateralis* | 3.50±1.80 (n=55) | 3.90±1.80 (n=19) | 3.25±1.38 (n=20) | 0.57 |
| ***Biochemical data*** | | | | |
| HbA1c (%) | 6.66±1.03 | 6.88±1.23 | 6.38±0.84 | 0.10 |
| FPG (mmol/L) | 7.0±1.8 | 7.3±2.5 | 6.8±1.5 | 0.38 |
| Hemoglobin (mmol/L) | 9.23±0.77 | 9.02±0.79 | 9.38±0.83 | 0.13 |
| Creatinine (µmol/L) | 82.23±27.41 | 96.38±38.9 | 76.04±11.49 | **0.02** |
| eGFR (ml/min/1.73m$^2$) | 85.68±18.03 | 73.96±19.96 | 92.84±11.65 | **<0.001** |
| Total cholesterol (mmol/L) | 3.66 ±0.91 | 3.65±0.96 | 3.82±0.94 | 0.52 |
| HDL (mmol/L) | 1.24±0.27 | 1.27±0.29 | 1.20±0.28 | 0.36 |
| LDL (mmol/L) | 1.79±0.83 | 1.67±0.82 | 2.09±0.81 | 0.08 |
| Triglycerides (mmol/L) | 1.37 ±0.77 | 1.57±1.06 | 1.17±0.49 | 0.09 |
| HOMA-IR* | 5.65±6.00 | 7.06±6.81 | 5.53±5.62 | 0.50 |

SBP: Systolic blood pressure; DBP: Diastolic blood pressure; FPG: Fasting plasma glucose; eGFR: Estimated glomerular filtration rate; HDL: High-density lipoprotein; LDL: Low-density lipoprotein.

Significance level was set at p<0.05.

*Data not normally distributed are presented as median±IQR; Mann–Whitney U test was used.

**Table 2. Results of the cardiopulmonary exercise test, near-infrared spectroscopy and echocardiography.**

| Based on Gläser (2010) | Total (N = 74) | Lowest fitness (N = 25) | Highest fitness (N = 25) | p value |
|---|---|---|---|---|
| *CPET data* | | | | |
| Rest V0$_2$ (mL/kg/min) | 4.44 ± 0.92 | 4.45 ± 0.76 | 4.56 ± 0.91 | 0.63 |
| V0$_2$ @ VAT (mL/kg/min) | 14.26 ± 3.94 | 11.06 ± 2.08 | 17.76 ± 3.59 | **<0.001** |
| Peak V0$_2$ (mL/min) | 2137.87 ± 701.53 | 1586.48 ± 408.78 | 2796.28 ± 643.35 | **<0.001** |
| Peak V0$_2$ (mL/kg/min) | 23.64 ± 6.68 | 17.72 ± 2.80 | 30.25 ± 5.67 | **<0.001** |
| Predicted Peak V0$_2$ - % (Glaser – 2010) | 96.68 ± 19.78 | 76.96 ± 10.57 | 118.44 ± 12.50 | **<0.001** |
| Peak workload (watt) | 186.19 ± 65.84 | 132.08 ± 36.21 | 246.48 ± 56.89 | **<0.001** |
| Peak HR (bpm) | 145.14 ± 26.76 | 125.68 ± 27.31 | 162.48 ± 15.17 | **<0.001** |
| Peak ventilation (L/min) | 85.59 ± 26.00 | 64.98 ± 17.96 | 108.46 ± 23.10 | **<0.001** |
| Peak RER | 1.16 ± 0.08 | 1.14 ± 0.10 | 1.18 ± 0.06 | 0.10 |
| Peak RPE | 16.51 ± 2.08 | 16.52 ± 2.31 | 16.44 ± 2.13 | 0.90 |
| VE/VCO$_2$ slope | 29.19 ± 4.45 | 31.18 ± 4.49 | 26.86 ± 2.84 | **<0.001** |
| V0$_2$/watt slope | 9.99 ± 1.41 | 9.82 ± 1.33 | 10.01 ± 0.98 | 0.57 |
| **NIRS data** | | | | |
| *During Exercise* | | | | |
| TSI baseline | 59.96 ± 4.89 | 60.65 ± 5.62 | 59.48 ± 5.26 | 0.45 |
| TSI max | 63.25 ± 4.57 | 63.60 ± 5.41 | 63.04 ± 4.19 | 0.68 |
| TSI min | 50.72 ± 6.69 | 50.10 ± 8.96 | 50.76 ± 4.67 | 0.74 |
| ΔTSI | 12.53 ± 5.53 | 13.50 ± 7.82 | 12.28 ± 3.36 | 0.47 |
| ΔHHb | 12.67 ± 5.93 | 11.42 ± 7.16 | 14.51 ± 4.02 | 0.07 |
| ΔO$_2$Hb | 7.20 ± 3.50 | 7.65 ± 3.69 | 7.35 ± 2.97 | 0.75 |
| ΔtHb | 13.29 ± 5.48 | 10.89 ± 4.70 | 16.66 ± 4.73 | **<0.001** |
| ΔHb$_{diff}$ | 15.27 ± 8.73 | 15.43 ± 11.46 | 15.64 ± 4.84 | 0.93 |
| ΔHHb/ΔtHb | 0.99 ± 0.37 | 1.05 ± 0.51 | 0.90 ± 0.21 | 0.19 |
| *During Recovery* | | | | |
| TSI min | 53.00 ± 6.07 | 52.88 ± 7.90 | 52.96 ± 5.17 | 0.97 |
| TSI max | 70.20 ± 3.34 | 70.09 ± 3.33 | 70.50 ± 2.70 | 0.63 |
| ΔTSI | 17.20 ± 6.03 | 17.22 ± 8.15 | 17.54 ± 3.99 | 0.86 |
| ΔHHb | 12.03 ± 6.00 | 10.74 ± 6.96 | 13.77 ± 4.51 | 0.07 |
| ΔO$_2$Hb | 17.70 ± 7.01 | 14.06 ± 7.25 | 21.32 ± 4.49 | **<0.001** |
| ΔtHb | 8.81 ± 3.32 | 6.61 ± 2.78 | 11.14 ± 2.91 | **<0.001** |
| ΔHb$_{diff}$ | 28.65 ± 12.54 | 24.13 ± 14.02 | 33.82 ± 8.36 | **0.005** |
| ΔHHb/ΔtHb | 1.79 ± 2.28 | 2.54 ± 3.70 | 1.39 ± 0.87 | 0.14 |
| *Echocardiography* | | | | |
| *Rest – lateral position* | | | | |
| Rest CO (L/min) | 5.93 ± 1.51 | 6.08 ± 1.97 | 6.01 ± 1.35 | 0.88 |
| Rest CI (L/min/m$^2$) | 2.89 ± 0.72 | 2.99 ± 1.02 | 2.86 ± 0.51 | 0.59 |
| *During exercise – semi supine position* | | | | |
| Rest CO (L/min) | 5.38 ± 1.31 | 5.62 ± 1.46 | 5.25 ± 1.48 | 0.47 |
| HR Rest (bpm) | 70.37 ± 11.33 | 73.35 ± 13.62 | 69.72 ± 9.72 | 0.37 |
| SV Rest (mL/min) | 77.06 ± 18.52 | 77.88 ± 21.55 | 74.84 ± 17.46 | 0.65 |
| Low CO (L/min) | 9.56 ± 2.25 | 8.85 ± 2.35 | 10.17 ± 2.42 | 0.11 |
| Peak CO (L/min) | 12.27 ± 3.00 | 10.26 ± 2.47 | 13.87 ± 3.18 | **<0.001** |
| Peak HR (bpm) | 131.79 ± 23.00 | 117.82 ± 22.18 | 144.72 ± 14.13 | **<0.001** |

*(Continued)*

**Table 2.** (Continued)

| Based on Gläser (2010) | Total (N = 74) | Lowest fitness (N = 25) | Highest fitness (N = 25) | p value |
|---|---|---|---|---|
| Peak SV (mL/min) | 94.52 ± 21.48 | 89.74 ± 24.17 | 96.44 ± 23.06 | 0.407 |
| Peak CI (L/min/m²) | 6.04 ± 1.40 | 5.10 ± 1.13 | 6.71 ± 1.18 | **<0.001** |

HR: Heart rate; RER: Respiratory exchange ratio; RPE: Rate of perceived exertion; VE/VCO$_2$: Ventilatory equivalent for CO$_2$; VO$_2$/watt: Oxygen consumption per watt; TSI: Tissue saturation index; HHb: Deoxygenated hemoglobin; O$_2$Hb: Oxygenated hemoglobin; tHb: Total hemoglobin; Hbdiff: O$_2$Hb − HHb; CO: Cardiac output; CI: Cardiac index.

The significance level was set at p < 0.05.

The average exercise capacity in our study was 23.64 ml O$_2$/kg/min, corresponding to 97% of the predicted VO$_2$peak. Participants in the lowest fitness group performed 22% below their predicted exercise capacity, whereas those in the highest fitness group exceeded their predicted values by 18% on average. While the difference between both groups was substantial, the average exercise capacity of our population was higher than that reported in previous studies investigating exercise capacity in adults with T2DM [10,11].

We categorized adults with T2DM in fitness groups based on their achieved VO$_2$peak, expressed as a percentage of predicted VO$_2$peak for their age, sex, length, height and smoking status. Despite this, the lowest fitness group was significantly older, suggesting that older adults with T2DM tend to perform worse relative to their healthy peers compared to younger adults with T2DM. This finding indicates that the impact of T2DM on fitness becomes more pronounced with advancing age. Furthermore, this group also had a longer history of diabetes, lower kidney function and was prescribed more cardiovascular drugs (i.e., diuretics and beta-blockers), which may reflect the cumulative effect of diabetes-related physiological maladaptations over time [1,26,27].

To better understand the physiological determinants underlying these fitness differences, we first compared the central and peripheral components between both fitness groups. Peak CO was significantly higher in the highest fitness group. This finding contrasts with previous studies that found no differences in peak CO between individuals with T2DM and exercise intolerance and individuals with T2DM but normal exercise capacity [10,11]. However, the lower peak CO observed in the lowest fitness group may be partially explained by their significantly lower maximal heart rate, potentially due to older age and more frequent use of beta-blockers [28,29].

When comparing the NIRS-derived outcome measures, a greater increase in tHb was found in the higher compared to the lower fitness group. Given that tHb serves as a marker for local tissue perfusion, the higher tHb in the highest fitness group might partly reflect the greater peak CO observed in these individuals [30,31]. However, patients in the lowest fitness group also tended to be more insulin resistant as shown by a greater HOMA-IR index and fasted plasma glucose, although not significantly different from the highest fitness group. It is well established that insulin resistance is associated with reduced capillary recruitment and endothelial dysfunction [32–36]. As such, individuals in the lower fitness group may have exhibited an impaired local muscle blood flow and vasodilatory response which could also have contributed to the lower tHb in this group.

This reduced muscle perfusion may explain the observed differences in O$_2$Hb during higher exercise intensities, as individuals with lower fitness may have had insufficient oxygen delivery to meet the increasing demand [37,38]. In contrast, those in the higher fitness group maintained O$_2$Hb levels close to baseline value, suggesting that oxygen delivery and demand were more effectively matched. This greater increase in muscle perfusion in the highest fitness group was accompanied by a parallel increase in HHb, suggesting not only greater perfusion but also more effective oxygen utilization at the muscular level [31,39]. However, it should be noted that HHb reflects the oxygen extraction, relative to the muscle perfusion [40]. Therefore, previous studies have recommended correcting for blood volume when assessing mitochondrial capacity [41,42]. Consequently, we introduced ΔHHb/ΔtHb as a volume-corrected marker of oxygen extraction

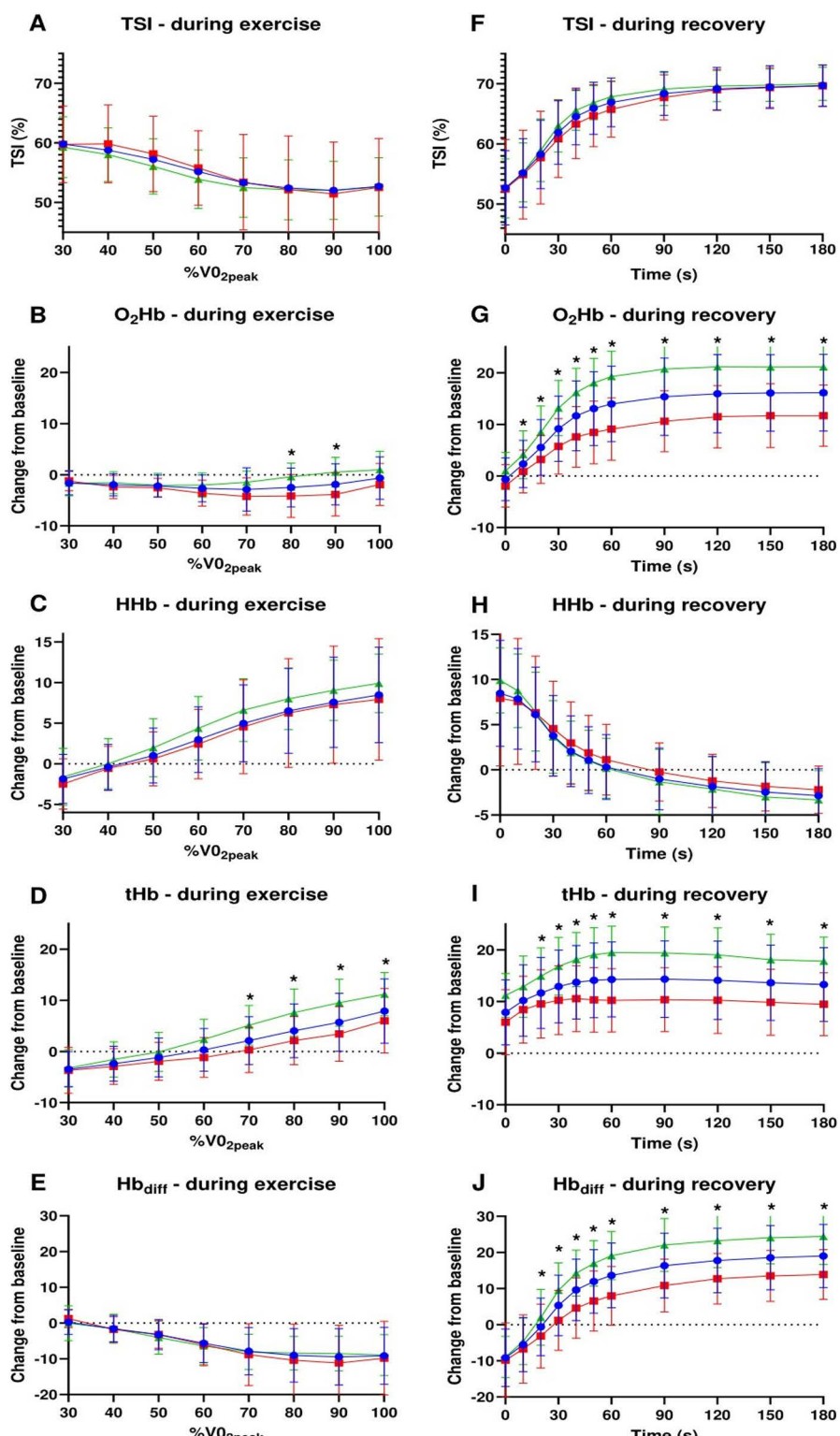

**Fig 1. Changes in NIRS-derived parameters during exercise and recovery.** TSI is expressed as a percentage. $O_2Hb$, HHb, tHb and $Hb_{diff}$ are expressed as changes relative to the start of the exercise. Blue circles represent average for the total sample, green triangles represent highest fitness group. Red squares represent lowest fitness group. Values are presented as means±SD. *: significant post-hoc analysis following repeated measures ANOVA ($P<0.05$).

**Table 3.** Overview of stepwise multivariate regression models on VO$_2$ peak (mL/min).

| Model | Covariates | Correlation | | Multivariate regression | | | | |
|---|---|---|---|---|---|---|---|---|
| | | r | p value | Adjusted R$^2$ | Standardised β | RMSE | BIC | p value |
| 1 | Age<br>Gender<br>Height<br>Weight<br>Smoking status | −0.51<br>−0.50<br>0.56<br>0.52<br>0.05 | <0.001<br><0.001<br><0.001<br><0.001<br>0.66 | 0.53 | | 483.41 | 1151.85 | <0.001 |
| 2 | Model 1<br>+ peak CO | 0.63 | <0.001 | 0.61 | 0.32 | 436.21 | 803.79 | <0.001 |
| 3 | Model 1<br>+ Peak HR<br>+ Peak SV | 0.43<br>0.29 | 0.001<br>0.035 | 0.68 | 0.49<br>0.21 | 398.94 | 797.28 | <0.001 |
| 4 | Model 1<br>+ ΔHHb/ΔtHb | −0.19 | 0.10 | 0.54 | −0.13 | 475.58 | 1149.43 | <0.001 |
| 5 | Model 1<br>+ peak CO<br>+ ΔHHb/ΔtHb | | | 0.62 | 0.31<br>−0.14 | 429.79 | 805.03 | <0.001 |

CO: Cardiac output; HR: heart rate; SV: stroke volume; ΔHHb/ΔtHb: change in deoxygenated hemoglobin divided by change in total hemoglobin; RMSE: Root mean square error; BIC: Bayesian information criterion.

Significance level was set at p<0.05.

capacity. As no differences in ΔHHb/ΔtHb were observed between both fitness groups, the differences in HHb may be attributable to increased muscle perfusion rather than to intrinsic differences in muscle oxygen extraction capacity.

The results of the multiple regression analysis further confirmed these findings, as the model including peak CO added significant explanatory value beyond traditional demographic and lifestyle factors (as described in the Gläser formula), including age, while ΔHHb/ΔtHb did not significantly correlate with VO$_2$peak and provided only minimal contribution to model performance. Although combining both peak CO and ΔHHb/ΔtHb provided a marginally better model fit, the improvement over the model with only peak CO was negligible, reinforcing peak CO as the dominant limiting factor in this population. This is in contrast with previous research where oxygen extraction capacity has been highlighted as a predictor for VO$_2$peak and exercise intolerance in adults with T2DM [11,43]. However, as previously mentioned, the population in this study was rather fit and therefore severe vasogenic remodeling might not have been present in these individuals.

Interestingly, during recovery, both groups showed a continued increase in tHb, suggesting that muscle blood flow may have been constrained during peak exercise. Previous research highlights that muscle contractions can indeed restrict blood flow by exerting pressure on the intramuscular capillaries [44]. While reports on post-exercise changes in tHb are limited, similar trends of continued tHb elevation during recovery have been previously observed [45,46]. These findings call for caution when interpreting changes in tHb as a direct reflection of CO, as maximal CO and maximal muscle perfusion might not be reached simultaneously.

### Strengths and limitations

The primary strength and novelty of our study is that both CO (as the central component) and NIRS-derived skeletal muscle hemodynamics (as the peripheral component) were assessed during exercise within the same participants, providing a more complete overview of the oxygen cascade.

However, the study has certain limitations. NIRS-derived skeletal muscle hemodynamics were measured at one single site on the vastus lateralis, which may limit the generalizability of the findings to the entire muscle or other muscle groups [47]. Given that local variations in muscle oxygenation have been previously documented, evaluating spatial heterogeneity would require the use of multi-channel NIRS equipment [47–49].

Despite the inclusion of a relatively large sample size, a significant proportion of NIRS measurements were of insufficient quality and could not be included in the analysis. These insufficient-quality NIRS-data were more prevalent among those with greater adipose tissue thickness, a factor known to affect NIRS-derived data [50,51]. This limitation is particularly relevant, as these individuals also exhibited a significantly lower exercise capacity. Consequently, the final study sample may not be fully representative of the overall population with T2DM. Moreover, while ANCOVA using either ATT or body fat percentage as covariates did not change the statistical significance of the findings between fitness groups, results should still be interpreted with caution. Although physiological calibrations, such as arterial occlusion, are often recommended to improve data interpretability, they are difficult to implement in this population [30,47,52]. Therefore, to improve the assessment of muscle hemodynamics in clinical populations, future studies could benefit from using NIRS devices with greater penetration depth to mitigate these limitations or from using alternative exercise protocols targeting muscle sites with less adipose tissue interference [48,52].

## Conclusion

Cardiac output was identified as the main determinant of $VO_2$peak, while differences in muscle oxygen extraction appeared to result primarily from variations in perfusion, rather than intrinsic limitations in mitochondrial function. The lower peak CO observed in the lower fitness group may be partly due to a reduced maximal heart rate, likely influenced by older age and more frequent use of CO-modulating medication. However, CO and local muscle perfusion are not directly equivalent, as mechanical factors, such as intramuscular pressure during high-intensity exercise, may prevent maximal perfusion.

## Supporting information

**S1 Table. Differences between low and high quality NIRS groups.**
(DOCX)

**S2 Table. Near-Infrared Spectroscopy results adjusted for adipose tissue thickness using ANCOVA.**
(DOCX)

**S3 Table. Near-Infrared Spectroscopy results adjusted for body fat using ANCOVA.**
(DOCX)

## Author contributions

**Conceptualization:** Matthijs Michielsen, Jomme Claes, Dominique Hansen, Guido Claessen, Marieke De Craemer, Véronique Cornelissen.

**Data curation:** Matthijs Michielsen, Louise Costalunga, Sara Amyay, Varvara Lazarou, Daphni Daraki, Eleftheria Kounalaki.

**Formal analysis:** Matthijs Michielsen, Panagiotis Chatzinikolaou.

**Funding acquisition:** Dominique Hansen, Guido Claessen, Marieke De Craemer, Véronique Cornelissen.

**Investigation:** Matthijs Michielsen, Youri Bekhuis, Elise Decorte, Camille De Wilde, Louise Costalunga, Sara Amyay, Varvara Lazarou, Daphni Daraki, Eleftheria Kounalaki, Kaatje Goetschalckx.

**Methodology:** Jomme Claes, Marieke De Craemer, Véronique Cornelissen.

**Resources:** Marieke De Craemer, Véronique Cornelissen.

**Supervision:** Jomme Claes, Marieke De Craemer, Véronique Cornelissen.

**Writing – original draft:** Matthijs Michielsen, Youri Bekhuis.

**Writing – review & editing:** Jomme Claes, Elise Decorte, Camille De Wilde, Tin Gojevic, Louise Costalunga, Sara Amyay, Varvara Lazarou, Daphni Daraki, Eleftheria Kounalaki, Panagiotis Chatzinikolaou, Kaatje Goetschalckx, Dominique Hansen, Guido Claessen, Marieke De Craemer, Véronique Cornelissen.

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
