## [Decision Letter · Decision Letter 0]

19 Jun 2025

PONE-D-25-24411Exploring the limits of exercise capacity in adults with type II diabetesPLOS ONE

Dear Dr. Michielsen,

Thank you for submitting your manuscript to PLOS ONE. After careful consideration, we feel that it has merit but does not fully meet PLOS ONE’s publication criteria as it currently stands. Therefore, we invite you to submit a revised version of the manuscript that addresses the points raised during the review process. I believe that the reviewers have provided very useful feedback and that by addressing their comments you will be able to improve the overall quality of the manuscript.

We look forward to receiving your revised manuscript.

Kind regards,

Juan M. Murias

Academic Editor

PLOS ONE

Journal Requirements:

2. Information for Editor:

Note from Nithya Chari (plosone@plos.org):

This manuscript reports a study of data collected during a clinical trial. The main trial is registered at https://clinicaltrials.gov. If a statistical review is needed, email plosone@plos.org and PLOS staff will assign one of our statistical advisors (http://journals.plos.org/plosone/s/advisory-groups#loc-statistical-advisors ).

This trial received funding from the Scientific Research Foundation of Flanders (FWO – T004420N and FWO – G095221N)

6. Please amend your list of authors on the manuscript to ensure that each author is linked to an affiliation. Authors’ affiliations should reflect the institution where the work was done (if authors moved subsequently, you can also list the new affiliation stating “current affiliation:….” as necessary).

Reviewers' comments:

Reviewer's Responses to Questions

**Comments to the Author**

1. Is the manuscript technically sound, and do the data support the conclusions?

Reviewer #1: Yes

Reviewer #2: Partly

2. Has the statistical analysis been performed appropriately and rigorously? 

Reviewer #1: Yes

Reviewer #2: Yes

3. Have the authors made all data underlying the findings in their manuscript fully available?

Reviewer #1: Yes

Reviewer #2: Yes

4. Is the manuscript presented in an intelligible fashion and written in standard English?

Reviewer #1: Yes

Reviewer #2: Yes

5. Review Comments to the Author

Reviewer #1: Title: EXPLORING THE LIMITS OF EXERCISE CAPACITY IN ADULTS WITH TYPE II DIABETES

The purpose of this study is “to explore these mechanisms by examining the relative contributions of CO and peripheral oxygen extraction to VO2peak. Specifically, we aim to determine whether differences in VO2peak between individuals with T2DM with the lowest and highest fitness are primarily driven by differences in CO, peripheral oxygen extraction or a combination of both.”

The outcomes of this study have the potential to advance the current understanding of the mechanisms leading to exercise intolerance in adults with type II diabetes. The study is methodologically well designed overall; however, I have some concerns I would like the authors to consider.

GENERAL COMMENTS

Concern 1: Groups not matched by age.

The age inclusion criteria for this study is very large (35-80 years old), potentially affecting the aerobic fitness comparison between low fitness and high fitness. Indeed, as reported in the Result section, there was a significant difference in age between low versus high fitness groups. Could the lower outcomes in VO2peak, cardiac output (CO), and NIRS-derived muscle oxygenation in the low fitness group have been significantly influenced by age? To what extent age alone would contribute to these findings? Considering the wide range of ages, the Introduction should give some information on how age affects the pathophysiology of type II Diabetes, and this should connect to the purpose of the study (see specific comments below).

I believe this is an important limitation of this study that will need to be addressed.

Concern 2: NIRS assessment and interpretation.

Measuring the amplitude changes of NIRS parameters always comes with limitations for between group comparisons. A physiological calibration as described by Barstow (2019), would have helped to limit signal contamination due to adipose tissue thickness for continuous wave NIRS devices, such as the one used in this study.

SPECIFIC COMMENTS

In the Introduction, I would suggest adding more information about the effect of age on type II diabetes and on the physiological responses to exercise. If the authors agree, the purpose of the study should also change accordingly to reflect the population being recruited. Age alone can affect both central and peripheral cardiovascular responses to exercise and the authors, in my opinion, should better link their study to the population being investigated.

Lines 9 – 11: I think this portion about the Fick equation should be placed at the beginning of the following paragraph. In the first paragraph it comes a bit out of nowhere.

Lines 29 – 33: again, in these lines and in general in the entire Introduction section it seems that the authors will compare two matched groups, which is not the case considering that age was not accounted for. I think this should be changed accordingly.

Lines 62- 63: Can the authors add the range of work rates increments used for the incremental test?

Similar to the Introduction, the Discussion section should address the effect of age a little bit deeper.

Additionally, I think that limitations regarding the NIRS discussed above should also be addressed.

Lines 200 – 204: the authors stated: “In contrast, those with in the higher fitness group maintained O2Hb levels close to baseline value, suggesting that oxygen delivery and demand were more effectively matched. Consequently, this significant greater increased in muscle perfusion is also reflected in a trend towards greater increases in HHb, a marker of oxygen consumption by the muscles, although this was not significant (29,36).”

This statement is controversial in my opinion. I don’t think that increased muscle perfusion reflects greater HHb (i.e., muscle oxygen extraction). Theoretically it should be the opposite. According to the Fick equation, greater perfusion should decrease HHb. What I see here is that both greater muscle perfusion and the intracellular capacity to utilize oxygen are optimized in the higher fitness group. Greater muscle perfusion cannot "consequently" reflect greater muscle oxygen extraction (i.e., HHb). Muscle oxygen extraction (HHb) reflects the balance between oxygen delivery and utilization (Grassi and Quaresima, 2016). I suggest the authors to reformulate this section.

Reviewer #2: I commend the authors for conducting a difficult study using challenging NIRS interpretations in a pathological condition such as T2DM during exercise. It is impressive to see the sample size and it is unfortunate, I believe, that no other NIRS device or methodology was used to carry out these potentially intriguing measurements. I am, however, confused about the aims/objectives and rationale of this study and I am a bit disappointed in the fact that existing literature about the application of NIRS in clinical populations with high subcutaneous adipose tissue thickness was not included nor known before the execution of this project (Ferrari et al. 2011, Hamaoka et al, 2011 and Celie et al., 2016). In general, I am/was a bit lost in the aims – scientific story you want to tell in the current study presented. Is it the link between central and local muscle perfusion? Is it an analysis of all parameters of the Fick equation, including muscle O2 extraction and consumption (in the mitochondria)? Is it the impact of B blockers (and diuretics) on central and peripheral haemodynamics? Is it something else? I think the authors should review the storyline and address this issue.

- In the study objectives, it was written that: ‘This study aims to explore these mechanisms by examining the relative contributions of CO and peripheral oxygen extraction to VO2peak. Specifically, we aim to determine whether differences in VO2peak between individuals with T2DM with the lowest and highest fitness are primarily driven by differences in CO, peripheral oxygen extraction or a combination of both. ‘ In the results nor discussion section, however, I haven’t really found how you handled, calculated, analysed O2 extraction. It seems to me that the authors predominantly focused on the links between central and local heamodynamics or blood flow (CO and muscle perfusion measured by tHb) as it is clearly written accordingly in the conclusion. Hence, I think this manuscript should be rewritten according to clearly defined objectives that are in agreement with the results – discussion and conclusions. I don’t really think that you presented the Fick equation data with regard to a-v O2 difference.

- An important remark and point I want to make is that I think it is crucial to include an analysis about the correlation/regression between subcutaneous adipose tissue and tHb to investigate how big the impact was of anthropometric data on tHb signal or amplitude presented. Do the authors observe a ‘real’ predictive muscle perfusion effect on VO2peak or could it be explained by stronger or weaker amplitudes due to higher or lower subcutaneous adipose tissue. Despite the fact that this point was included in strengths and limitations, I am interested in what the data says (Vastus Lateralis Adipose tissue and tHb data are available.)

- Considering the fact that B-blockers could also affect muscle perfusion, I was wondering whether no collinearity was found between both parameters in the regression analysis. Are these factors linked to each other? Did you also investigate homo- and heteroscedasticity of the regression data?

- Did the authors investigate the impact of diuretics on VO2 peak, CO and muscle perfusion? Taking into consideration the significant different intake between the higher and lower fitness group, it might be interesting to investigate and include in the regression analysis.

- Hence, as the authors wrote in the conclusion, it is quite relevant to see that B-blockers (and maybe diuretics too) hamper VO2 peak through impacting muscle perfusion, CO and VO2 peak. The regression model present these factors as independent from each other (B-blocker and tHb were included as independent predictors of VO2 peak, but are they independent…?)

- Is there data available about the degree of insulin resistance in both the higher and lower fitness groups? Was HOMA-IR available?

In conclusion, I think the authors need to rethink the presentation or the angle of their data/study and align it with clearly defined aims, results, discussion and conclusion. Notwithstanding my critical view towards the current form of the study, I believe that it does, however contain relevant information to publish.

Minor remarks:

Line 2-5: Adjust sentence, not correct

Line 177-179: Eliminate Fick equation as muscle perfusion rather than O2 extraction is reported.

Line 223-226 : Here the differences in subcutaneous adipose tissue could be a crucial mediator as well… Did you measure real muscle oxy and deoxygenation or lipied oxygenation dynamics?

6. PLOS authors have the option to publish the peer review history of their article (what does this mean? ). If published, this will include your full peer review and any attached files.

**Do you want your identity to be public for this peer review?** For information about this choice, including consent withdrawal, please see our Privacy Policy .

Reviewer #1: No

Reviewer #2: No

---

## [Author Response · Author response to Decision Letter 1]

3 Aug 2025

We would like to thank the reviewers for taking the time to carefully read and critically evaluate our manuscript. The constructive comments and suggestions have significantly helped us to improve the quality of the work.

As requested, a point-by-point response to the reviewers’ comments is provided

---

## [Decision Letter · Decision Letter 1]

21 Aug 2025

Exploring the limits of exercise capacity in adults with type II diabetes

PONE-D-25-24411R1

Dear Dr. Michielsen,

We’re pleased to inform you that your manuscript has been judged scientifically suitable for publication and will be formally accepted for publication once it meets all outstanding technical requirements.

Kind regards,

Juan M. Murias

Academic Editor

PLOS ONE

Additional Editor Comments (optional):

Reviewers' comments:

Reviewer's Responses to Questions

**Comments to the Author**

1. If the authors have adequately addressed your comments raised in a previous round of review and you feel that this manuscript is now acceptable for publication, you may indicate that here to bypass the “Comments to the Author” section, enter your conflict of interest statement in the “Confidential to Editor” section, and submit your "Accept" recommendation.

Reviewer #1: All comments have been addressed

Reviewer #2: All comments have been addressed

2. Is the manuscript technically sound, and do the data support the conclusions?

Reviewer #1: Yes

Reviewer #2: Yes

3. Has the statistical analysis been performed appropriately and rigorously? 

Reviewer #1: Yes

Reviewer #2: Yes

4. Have the authors made all data underlying the findings in their manuscript fully available?

Reviewer #1: Yes

Reviewer #2: Yes

5. Is the manuscript presented in an intelligible fashion and written in standard English?

Reviewer #1: Yes

Reviewer #2: Yes

6. Review Comments to the Author

Reviewer #1: I thank the authors for their revisions and for addressing my concerns. The quality of the manuscript has improved following the authors revisions.

Reviewer #2: Dear authors,

I am happy with the made adjustments and I believe the manuscript has improved a lot in terms of storyline, analyses and the 'treatment or presentation' of the NIRS data which is a difficult physiological measurement to interpret and especially challenging in clinical populations. My final and small remark was whether the authors could please revise the very first sentence of the introduction, which does contain a linguistic error or is incomprehensible to me.

7. PLOS authors have the option to publish the peer review history of their article (what does this mean? ). If published, this will include your full peer review and any attached files.

**Do you want your identity to be public for this peer review?** For information about this choice, including consent withdrawal, please see our Privacy Policy .

Reviewer #1: No

Reviewer #2: No

---

## [Editor Report · Acceptance letter]

PONE-D-25-24411R1

PLOS ONE

Dear Dr. Michielsen,

I'm pleased to inform you that your manuscript has been deemed suitable for publication in PLOS ONE. Congratulations! Your manuscript is now being handed over to our production team.

Kind regards,

on behalf of

Dr. Juan M. Murias

Academic Editor

PLOS ONE